# Amino Acid-Derived Bacterial Metabolites in the Colorectal Luminal Fluid: Effects on Microbial Communication, Metabolism, Physiology, and Growth

**DOI:** 10.3390/microorganisms11051317

**Published:** 2023-05-17

**Authors:** François Blachier

**Affiliations:** Université Paris-Saclay, AgroParisTech, INRAe, UMR PNCA, 91120 Palaiseau, France; francois.blachier@agroparistech.fr

**Keywords:** amino acids, intestinal bacterial metabolites, colorectal fluid, microbial communication, microbial physiology

## Abstract

Undigested dietary and endogenous proteins, as well as unabsorbed amino acids, can move from the terminal part of the ileum into the large intestine, where they meet a dense microbial population. Exfoliated cells and mucus released from the large intestine epithelium also supply nitrogenous material to this microbial population. The bacteria in the large intestine luminal fluid release amino acids from the available proteins, and amino acids are then used for bacterial protein synthesis, energy production, and in other various catabolic pathways. The resulting metabolic intermediaries and end products can then accumulate in the colorectal fluid, and their concentrations appear to depend on different parameters, including microbiota composition and metabolic activity, substrate availability, and the capacity of absorptive colonocytes to absorb these metabolites. The aim of the present review is to present how amino acid-derived bacterial metabolites can affect microbial communication between both commensal and pathogenic microorganisms, as well as their metabolism, physiology, and growth.

## 1. Introduction

The process of protein digestion in the small intestine is efficient, with a yield equal to or even above 90% for most alimentary proteins [1]. Following protein digestion in the small intestinal luminal fluid by pancreatic proteases, peptides are further degraded by epithelial peptidases, and released oligopeptides and amino acids are finally absorbed in the portal blood through the small intestine epithelium [2,3,4].

A minor part of undigested or not fully digested proteins, together with other nitrogenous compounds, can move from the terminal part of the ileum to the large intestine. Indeed, it has been estimated in volunteers that between 1.5 and 5 g of nitrogen are transferred every day through the ileo-caecal junction [5,6,7,8,9], with the major parts of this nitrogenous material being proteins and, to a much lesser extent, peptides, while the minor parts of this nitrogenous material are made of amino acids, urea, ammonia, nitrate, and other compounds [8]. Regarding peptides, it has been shown that these compounds are more rapidly absorbed in the small intestine than free amino acids [10,11]. Of note, the nitrogenous material recovered in the distal small intestine is not only originating from the diet but also from endogenous sources, including proteins present in the exocrine secretion, in the fully mature exfoliated enterocytes, and in the mucus layer released in the luminal fluid [12]. It has been determined that roughly 40% of the nitrogenous material transferred to the large intestine is from alimentation, while the remaining 60% is of endogenous origin [9].

By using a conversion factor between nitrogen and protein equal to 6.25 [13], it can be calculated as an approximation that roughly between 4 and 12 g of alimentary proteins escape digestion in the small intestine. This estimation fits rather well with the mean protein consumption in Western countries, which averages 85 g per day [14,15]. Indeed, if we assume a mean protein digestibility efficiency of approximately 90% [16], we can calculate that 8 g of alimentary proteins would be recovered in the large intestine, a value within the 4–12 g range as calculated above.

In addition to the nitrogenous material originating from the small intestine, exfoliation of fully mature colonic epithelial cells and mucus renewal in the large intestine may also contribute to the global amount of nitrogenous compounds available for the microbial population present in the colorectal fluid (Figure 1). In the large intestine luminal fluid, this nitrogenous material meets a dense population of microbes in a context of extended transit [17]. A part of the water in the luminal fluid contained within the large intestine is progressively absorbed by the epithelial cells from the proximal to the distal part, resulting in a median value of water content in the human fecal material equal to 75% of the total mass [18].

The gut microbiota is composed of a vast population of bacteria but also of archaea [19], viruses [20,21], and fungi [22]. Protozoans in the intestine, although classically not included as part of the microbiota itself, represent a heterogeneous group of eukaryotic organisms, with some of them being considered as parasites [23]. The term gut ecosystem usually refers to the biological community of microorganisms living in the exogenous environment of the gut. The metabolism of proteins and amino acids by the intestinal microbes has been mostly studied for the bacterial part of the intestinal microbiota [24,25], and as explained in the following paragraph, these compounds are used in bacteria in both anabolic and catabolic pathways.

## 2. Bacteria in the Large Intestinal Luminal Fluid Release Amino Acids from Undigested Proteins and Used Them for Protein Synthesis, Energy Production, and Other Catabolic Pathways Which Release Various Bacterial Metabolites

In the luminal fluid of the large intestine, proteins and peptides are degraded by the bacterial proteases and peptidases, then released as amino acids [26,27,28]. The proteolytic activities in the large intestine have been mainly attributed to the genera *Bacteroides*, *Clostridium*, *Propionibacterium*, *Fusobacterium*, *Streptococcus*, and *Lactobacillus* [29]. Some bacteria, such as lactic acid bacteria, have developed proteolytic systems, presumably to compensate for their reduced or even absent capabilities to synthesize amino acids [30]. These proteolytic systems include extracellular proteases that degrade proteins into oligopeptides, and this degradation is followed by the entry of oligopeptides within bacteria via dedicated transporters. Furthermore, intracellular peptidases degrade the peptides into amino acids [31,32]. Amino acids and their corresponding metabolites can be imported and exported from the bacteria via transmembrane proteins [33,34], allowing regulation of the concentration of these compounds in the bacterial cells and allowing exchanges of amino acids and related metabolites between bacterial species. Efflux systems for some amino acids such as lysine, arginine, threonine, cysteine, leucine, isoleucine, and valine have been studied in the bacteria *E. coli* and *Corynebacterium glutamicum* [35,36].

In the large intestine, amino acids resulting from the transfer of amino acids from the small to the large intestine, together with amino acids released from proteins and peptides that have not been digested in the small intestine and amino acids released from endogenous proteins and peptides in the large intestine, are used by the intestinal microbiota for their metabolism. Amino acid availability for microbial metabolism in the large intestine is presumably little decreased by amino acid absorption through the colonocytes since, except in the neonatal period [37,38], the colonic mucosa does not absorb amino acids to any significant extent [39]. However, several amino acid transporters, as well as the peptide transporter PepT1 have been identified in the large intestine epithelium [40], suggesting that small amounts of amino acids would be transported inside and/or through colonocytes [39].

Since approximately half of the bacterial biomass in the colon is lost every day by defecation [41], the rapid growth of bacteria within the large intestine luminal fluid requires extensive amounts of substrates, including amino acids, that are mainly provided by the host. This intense bacterial metabolism and associated anabolism are possible only if the required substrates are provided in sufficient quantities [24]. Many metabolic pathways involved in amino acid synthesis in bacteria are conserved in the different bacterial lineages, including those bacteria that inhabit the large intestine [42,43]. However, major differences remain when comparing the metabolic capacities for amino acid synthesis at the level of species and strains. As presented above, some bacteria, such as lactic acid bacteria, possess low or even absent capacities for amino acid biosynthesis. Another illustrative example of such bacteria is represented by *Clostridium perfringens*, which displays no metabolic capacity for the synthesis of glutamate, arginine, histidine, lysine, methionine, serine, threonine, aromatic, and branched-chain amino acids [44], thus depending on the host for the supply of these amino acids for their metabolic needs. *Lactobacillus johnsonii* is another gut bacterium that is unable to perform the synthesis of almost all amino acids due to the lack of numerous anabolic pathways required for amino acid synthesis [45]. Other intestinal bacteria, including *Enterococcus faecalis* and *Streptococcus agalactiae*, perform the synthesis of only a few specific amino acids in mammals (including humans) [46]. In sharp contrast, other intestinal bacteria such as *Clostridium acetobutylicum* are equipped with a complete set of genes required for the biosynthesis of all amino acids [47]. However, it is important to keep in mind that the sole presence of genes implicated in one given amino acid synthesis within one given bacterium does not allow for a conclusion on the functionality of the corresponding metabolic pathways. A typical example of such limitation is given by the common gut bacterium *Lactococcus lactis*, for which the genes allowing the synthesis of the 20 amino acids have been identified but which requires supplementation with isoleucine, valine, leucine, histidine, methionine, and glutamate for growth since genes involved in the biosynthetic pathways corresponding to these amino acids have been demonstrated to be non-functional due to point mutations [48,49].

Bacteria may incorporate the available amino acids into proteins or may use them as energy substrates and in various catabolic pathways [50] (Figure 1). Most gut bacteria utilize amino acids and ammonia as preferred nitrogen sources, and amino acids such as glutamine, glutamate, asparagine, aspartate, lysine, serine, threonine, arginine, glycine, histidine, and branched-chain amino acids are preferred substrates for degradation by gut bacteria, with numerous products being formed, notably ammonia, short-chain fatty acids, branched-chain fatty acids, phenols, indoles, organic acids, amines, and various gaseous compounds [51].

In healthy adults, the distal part of the intestinal tract harbors large communities of obligate anaerobes [52,53]. In the virtual absence of oxygen or other suitable electron acceptors, only strict or facultative anaerobic bacteria, such as Clostridia and Fusobacteria, can utilize amino acids as energy sources, thus fermenting amino acids and producing mainly branched-chain fatty acids and ammonia [54], as well as short-chain fatty acids, hydrogen (H_2_), carbon dioxide (CO_2_), hydrogen sulfide (H_2_S), and various organic compounds [55,56] as listed above. Several mechanisms for amino acid degradation have been described in anaerobic bacteria, including the Strickland reaction that is operative in several proteolytic Clostridia. This latter reaction involves the coupled oxidation and reduction of amino acids to organic acids. Other fermentation pathways described in various Clostridia as well as in *Fusobacterium* spp. and *Acidaminococcus* spp. involve amino acids that act as electron donors or acceptors [57,58]. The genus *Clostridium* contains specific amino acid degradation pathways, such as B1-dependent aminomutases, selenium-containing oxidoreductases, and oxygen-sensitive 2-hydroxyacyl-CoA dehydratases [59]. Amino acids can also be decarboxylated, ultimately yielding biogenic amines and polyamines. Of note, luminal parameters such as pH can modulate the catalytic activity of different bacterial enzymes such as deaminases and decarboxylases, thus affecting the production of specific metabolic end products [55].

Some amino acids, depending on the bacterial species considered, may be utilized for specific metabolic pathways. For instance, *Clostridium stricklandii* preferentially uses threonine, arginine, and serine as carbon sources and for energy production, but little utilizes glutamate, aspartate, and aromatic amino acids for such purposes and uses lysine as fuel only in the stationary growth phase [59]. The ratio of available carbohydrates over proteins is central for fixing substrate utilization by the gut microbiota in different contexts of substrate availability [54], and in humans, higher availability of complex carbohydrates (like plant fibers and resistant starch) lowers the process of protein fermentation by the intestinal bacteria, as determined by measuring several amino acid-derived bacterial metabolites in feces [60,61,62]. In addition, when fermentable carbohydrates are abundant for intestinal bacteria, amino acids are mostly used for anabolic metabolism but little for energy production [63]. On the other hand, in a context of low availability of fermentable carbohydrates, several amino acids are used by intestinal bacteria for energy production, thus supporting bacteria growth [42,50,64]. Due to high carbohydrate fermentation in the proximal colon, there is a progressive decrease in carbohydrate availability in the more median and distal parts of the colon, thus resulting in higher protein degradation and amino acid utilization [65]. Differential use of substrates when comparing different bacterial species has been documented. For instance, most genera in the phylum Firmicutes preferentially use proteins among the available substrates for their metabolism [66].

Relatively few nutritional intervention studies in volunteers have examined the effects of increasing the amounts of proteins in the diet on the gut microbiota composition and metabolic activity. In several of these studies, the dietary protein intake between the groups of volunteers was not the sole parameter modified, since modifications of energy and/or fiber were noticed in these studies. These two latter parameters are known to affect the gut microbiota composition [67,68,69,70,71,72], rendering correct interpretation of the results obtained from these studies difficult. Three studies have used high-protein diets without modification of dietary fiber or energy intake [73,74,75]. In these latter studies, the high-protein diets were made isocaloric by decreasing the amount of carbohydrates in the diet. Under such conditions, these three studies found little change in the fecal and rectal-associated bacterial composition after several weeks of consumption of the high-protein diet. In contrast, in the three studies, marked changes in the amino acid-derived bacterial metabolites were recorded both in feces and urine, reinforcing the view that luminal substrate availability is central to fixing bacterial metabolite concentrations in the luminal fluids.

However, substrate availability is not the sole parameter that influences the concentrations of amino acid-derived bacterial metabolites in biological fluids. Microbial composition and its overall metabolic activity, as well as the absorption of bacterial metabolites through colonocytes, are also believed to influence the luminal concentrations of these compounds [24]. Incidentally, approximately half of the metabolic pathways used by the intestinal bacteria do not occur in the cells of the host, and the largest group of such metabolic pathways involves pathways related to amino acid metabolism [76]. Other physiological parameters, such as transit time, may also influence the concentrations of amino acid-derived metabolites in the colon. Indeed, longer transit times are associated with higher levels of protein fermentation [77,78].

Finally, as explained above, the utilization of the different amino acids by the intestinal bacteria produces numerous amino acid-derived metabolites that are detected in the different biological fluids [24], and several of them, as detailed in the next paragraph, have been shown to play a role in the biology of intestinal microbes.

Although the rate of amino acid catabolism by the bacteria of the large intestine represents a major parameter to determine the concentrations of several amino acid-derived bacterial metabolites in the colorectal fluid, the rate of bacterial amino acid synthesis also likely plays a role in fixing such concentrations. In fact, intestinal bacteria use bacterial metabolites that are produced during amino acid catabolism (such as ammonia and hydrogen sulfide) [42] in the process of amino acid synthesis (Figure 1).

## 3. Amino Acid-Derived Bacterial Metabolites Are Involved in the Biology of Intestinal Microbes

Several amino acid-derived bacterial metabolites have been shown to be implicated in microbial communication and in microbial metabolism, physiology, and growth. Of note, as presented below, these effects involve both commensal and pathogenic intestinal microorganisms.

### 3.1. Lactate, Formate, Succinate and Oxaloacetate

During the catabolism of amino acids, the intestinal bacteria produce numerous organic acids, including lactate, formate, succinate, and oxaloacetate [29,50]. Indeed, high-protein diet consumption increases the amounts of organic acids in the large intestine luminal content [79]. Of note, these organic acids are not exclusively produced by the intestinal microbiota from amino acids but also from carbohydrates [80,81,82]. Among these organic acids, lactate is used as carbon and an energy source by indigenous bacteria, including *Salmonella* and *Campylobacter* [83]. Formate, which is secreted by the pathogenic bacteria *Shigella flexneri*, has been shown to promote the expression of genes involved in their virulence [84]. Oxaloacetate, when produced by *Escherichia coli* helps the parasite *Entamoeba histolytica* to survive in the large intestine [85] (Figure 2).

This latter result is of notable importance given that this parasite can trigger a strong inflammatory response upon invasion of the colonic mucosa. In addition, this study shows that communication between bacterial species and parasites through dedicated metabolites may occur in the large intestine. Lastly, succinate produced by the gut microbiota has been shown to promote infection by *Clostridium difficile* after antibiotic treatment [86]. Furthermore, *Clostridium butyricum* appears to be able to prevent *Clostridium difficile* proliferation by diminishing the succinate concentration in the large intestine luminal content [87]. Interestingly, this latter decreased succinate concentration is apparently the net result of overall modifications of the microbiota’s metabolic activity [87].

### 3.2. p-Cresol

The metabolite *p*-cresol (4-methylphenol) is produced by the intestinal microbiota of the large intestine from the amino acid tyrosine [88]. From in vitro analysis, it has been shown that among the bacteria present in the human gut, specific families of bacteria, namely *Fusobacteriaceae*, *Enterobacteriaceae*, *Clostridium*, and *Coriobacteriaceae* are strong *p*-cresol producers [89]. As expected, by increasing the protein content in the diet of mammals, an increased *p*-cresol concentration is measured in the feces [90]. On the contrary, the fecal excretion of *p*-cresol is diminished when the diet is enriched with undigestible polysaccharides [60].

Of major interest, the production of *p*-cresol by *Clostridium difficile* gives this bacterium a competitive advantage over other gut bacteria such as *Escherichia coli*, *Klebsiella oxytoca*, and *Bacteroides thetaiomicron* [91,92]. By using a model of *Clostridium difficile* infection in rodents, it has been demonstrated that excessive *p*-cresol production affects the gut microbiota diversity [91]. Furthermore, by removing the capacity of *Clostridium difficile* to produce *p*-cresol, the capacity of this bacterium to recolonize the intestine after an initial infection is diminished [91] (Figure 2). These results are carrying potential applications as *Clostridium difficile* is a major cause of diarrhea and inflammation in patients following long-term antibiotic treatment [93].

### 3.3. Indole

Indole is produced from tryptophan by various Gram-positive and Gram-negative species, including *Escherichia coli*, *Proteus vulgaris*, *Clostridium* spp. and *Bacteroides* spp. [94,95,96]. Indole diminishes cell motility and aggregation in *L. monocytogenes* [97] (Figure 2). Indole diminishes the virulence of *Pseudomonas aeroginosa* [98], of *Salmonella enterica* [99], and the virulence and growth of the fungal species *Candida albicans* [100]. In another study, indole was found to be bacteriostatic against lactic acid bacteria while affecting their survival [101]. Of note, indole mitigates cytotoxicity by *Klebsellia* spp. by suppressing toxin production by this bacterium [102]. In addition, indole impairs the ability of enteropathogenic *Escherichia coli* to enhance its virulence activity in response to *Vibrio cholerae* [103]. Lastly, *Clostridium difficile*, which itself is not considered an indole producer, increases indole production by other gut bacteria [104]. By doing so, *Clostridium difficile* increases the concentration of indole within the intestinal fluid, then limiting growth of indole-sensitive bacteria, and finally adversely improving *Clostridium difficile* growth and persistence within the intestinal luminal content [104].

Thus, if there is little doubt that indole is involved in intestinal microbe physiology and growth, the beneficial versus deleterious effects of this bacterial metabolite for gut health appears to depend on the overall colorectal ecosystem characteristics.

### 3.4. Skatole

Skatole is produced by the intestinal microbiota from tryptophan [54]. This bacterial metabolite has been shown to display a marked inhibitory effect on enterohemorrhagic *Escherichia coli* biofilm formation [105] (Figure 2). Briefly, biofilms are structures formed by the colon microbiota that are in contact with the mucosal surface and that play a role in the modulation of epithelial barrier function [106].

### 3.5. Hydrogen Sulfide

Hydrogen sulfide (H_2_S) is produced by numerous bacterial species in the large intestine from both dietary and endogenous S-containing substrates [107]. These substrates include cysteine, but also sulfate, taurine, and sulfomucins [108,109,110,111]. By using in vitro tests with human fecal material, it has been found that, for instance, production of H_2_S is greatly enhanced by sulfur-containing amino acids but much more modestly by inorganic sulfate [112]. In addition, H_2_S production is diminished by fermentable fibers [112], and there is a positive relationship between dietary protein intake and H_2_S production by the intestinal microbiota [113,114].

The protective role of H_2_S in bacteria was suggested more than six decades ago. Indeed, H_2_S produced by *Desulfovibrio desulfuricans* was demonstrated to be the diffusible factor responsible for protecting *Pseudomonas aeruginosa* from heavy metal toxicity (for instance, mercury) [115]. Similarly, H_2_S produced by *Escherichia coli* contributes to the protection of *Staphilococcus aureus* against mercuric chloride toxicity [116]. Much more recently, and in the context of the study of resistance to antibiotics, H_2_S has been shown to be a protective agent in bacteria such as *Pseudomonas aeruginosa* (found in human fecal samples [117]), *Staphylococcus aureus* (found in human intestine [118]), and *Escherichia coli* against the action of numerous antibiotics [119,120] (Figure 2). Sequestration of Fe^2+^ ions by H_2_S counteracts oxidative stress triggered by antibiotics in *Escherichia coli* [121]. Furthermore, H_2_S has been shown to be involved in the maintenance of redox homeostasis in bacteria and to protect bacteria against the oxidative stress triggered by the antibiotic ampicillin [122]. Also of major importance, cystathionine-Ɣ-lyase has been identified as the primary enzymatic activity that generates H_2_S in *Staphylococcus aureus* and *Pseudomonas aeruginosa*, and inhibition of this activity potentiates the efficiency of bactericidal antibiotics against both pathogens in in vitro and in vivo models of infection [123].

However, H_2_S is apparently not a bacterial metabolite that limits the efficiency of antibiotics against all pathogenic bacteria. Indeed, for instance, the pathogenic bacteria *Acinetobacter baumannii*, which is found in the intestinal tract [124] and does not produce H_2_S, is sensitive to exogenous H_2_S since, in these bacteria, this compound reinforces the effects of several classes of antibiotics [125]. Thus, the H_2_S-mediated protection or sensitization of intestinal bacteria to the bactericidal effects of several antibiotics apparently depends on the bacterial species examined.

Lastly, in another context, the implication of H_2_S in the resistance to infection by pathogenic bacteria has been recently suggested [126]. Indeed, in this latter study, the production of sulfide from taurine appears to be involved in the enhanced capacity of commensal bacteria to counteract pathogen infection.

### 3.6. Polyamines

Polyamines are small aliphatic amines that are produced by the intestinal microbiota from amino acids. The precursors for putrescine, spermidine, and spermine synthesis in bacteria are ornithine, arginine, and methionine, respectively, while agmatine is produced from arginine and cadaverine is produced from lysine [127]. The polyamines putrescine, spermidine, and spermine play a central role in bacterial growth [128] (Figure 2).

Regarding cadaverine, experimental works suggest that this polyamine plays a major role in the pathogenesis of *Shigella* infections [129]. In addition, cadaverine can prevent the escape of *Shigella flexneri* from the phagolysosome, and such an effect likely represents a way to connect the control of bacterial dissemination and neutrophil transepithelial signaling [130].

The polyamines agmatine and spermidine have been shown to be implicated in biofilm formation [131], while spermine inhibits *Vibrio cholerae* biofilm formation [132]. Spermidine reinforces the production of the bacterial genotoxin colibactin [133]. Lastly, one study reported that putrescine, cadaverine, spermidine, and spermine can modulate *Vibrio cholerae* virulence properties [134].

### 3.7. Gamma-Amino Butyric Acid, Norepinephrine, and Serotonin

Several bacterial species present in the mammalian gut, including the human gut, produce metabolites such as gamma-amino butyric acid, norepinephrine, and serotonin from different amino acids. These bacterial metabolites are well known to also be produced by the host with neurotransmitter functions [24]. There are emerging data that indicate that these metabolites are involved in the adaptation of bacteria to changes in their environment as well as in the modulation of bacterial physiology and growth. This is an exciting area of research, as it strongly suggests a very long evolutionary history for the functions of these compounds in the living world.

Gamma-amino butyric acid (GABA) is produced from glutamate by some intestinal bacteria, including strains of *Lactobacillus* and *Bifidobacterium*, as well as *Bacteroides* spp. [135,136,137]. GABA has been shown to be one component involved in bacterial acid tolerance in the context of changing luminal pH through the maintenance of intracellular pH [136,138,139] (Figure 2).

Regarding norepinephrine (also called noradrenaline), this bacterial metabolite is produced by the intestinal microbiota from tyrosine by several bacterial species, such as *Bacillus subtilis*, *Escherichia coli*, and *Proteus vulgaris* [140]. This bacterial metabolite affects the growth, either positively or negatively, depending on the bacterial species considered, of some anaerobic bacteria such as *Fusobacterium nucleatum*, *Prevotella* spp., *Klebsiella pneumoniae*, *Pseudomonas aeruginosa*, *Enterobacter clocae*, *Shigella sonnei*, and *Staphylococcus aureus*. Norepinephrine increases the virulence of several anaerobic bacteria, such as *Clostridium perfringens* [141,142,143].

Serotonin (5-hydroxy trypyamine) is produced from tryptophan by a vast number of bacterial species, among which *Escherichia coli*, *Bacteroides*, *Streptococcus*, *Bifidobacterium*, *Lactococcus*, *Lactobacillus*, and *Propionibacterium* [140]. Serotonin has been shown to promote the colonization of *Turicibacter sanguinis* in the human gut [144].

### 3.8. 4-Hydroxyphenylacetate

The bacterial metabolite 4-hydroxyphenylacetate (HPA) is a metabolic intermediary that is produced from tyrosine by phenol and *p*-cresol producers [89]. This compound inhibits the growth of the foodborne pathogen *Listeria monocytogenes* (Figure 2), an effect associated with alteration of the morphology of the bacteria and decreased expression of several virulence genes [145].

## 4. Conclusions and Perspectives

As presented in the present review, the results from experimental studies clearly indicate that numerous metabolites produced from amino acids of dietary and endogenous origin by the intestinal microbiota, which are found in stools and in the luminal fluid of the large intestine, are biologically active on microbes. The effects recorded are related to communication between microbes and to the physiology, metabolism, and growth of intestinal microorganisms. From the recorded effects of these bacterial metabolites on intestinal microbe biology, it is tempting to propose that the production of these compounds is devoted in the very first place to the dialogue between microorganisms coexisting in the mammalian intestinal colorectal luminal fluid.

However, the data obtained from experimental works are often characterized by several limitations that need to be taken into consideration for correct interpretation and future work. Firstly, the concentrations of the bacterial metabolites tested are not necessarily within the range of concentrations that are present in the vicinity of microbes in the colorectal fluid. Since this parameter is experimentally difficult to measure, the bacterial concentrations used in experiments often refer to the concentrations measured in feces, which represents an approximation of the concentrations within the rectal fluid but is likely different from those in the different segments of the colon [24]. Secondly, the intestinal luminal fluid contains a complex mixture of bacterial metabolites that can exert presumably additive, synergistic, or opposite effects on the intestinal microbial population. In fact, in most experimental works, the bacterial metabolites are tested individually, making it difficult to extrapolate from experimental works to “real-life situations”.

With these reservations in mind, the emerging experimental data indicate that intestinal bacteria produce several metabolites from amino acids that are involved in signaling between them and other microbial species (either commensal or pathogenic), as well as in the physiology and metabolism of these microorganisms, thus regulating their respective growth and biological activities. These data encourage clinical work in volunteers to test in different situations the relevance of dietary (or pharmacological) interventions for eventually controlling the colorectal microbial population and its metabolic activities in a way that would be beneficial for the host intestinal health from a preventive or curative perspective [24].

## Figures and Tables

**Figure 1 microorganisms-11-01317-f001:**
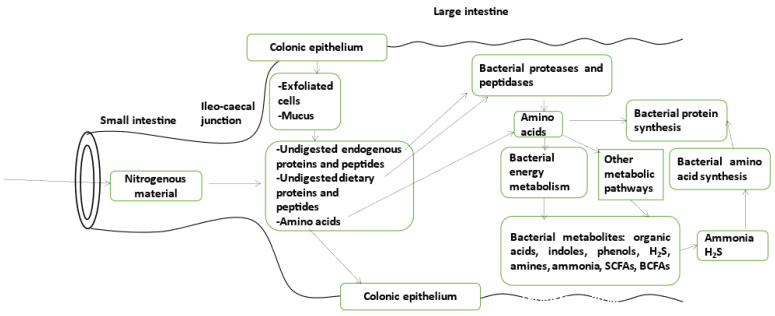
Schematic representation of the metabolic fate of undigested endogenous/dietary proteins and free amino acids in the large intestine luminal fluid. Undigested proteins are substrates for bacterial proteases and peptidases which release amino acids. Amino acids are little absorbed by the colonic epithelium, but are likely mostly used for bacterial protein synthesis, energy metabolism and other metabolic pathways, then releasing numerous bacterial metabolites. Several among these amino acid-derived metabolites have been shown to be active on the biology of microorganisms in terms of signaling, metabolism, physiology, and growth.

**Figure 2 microorganisms-11-01317-f002:**
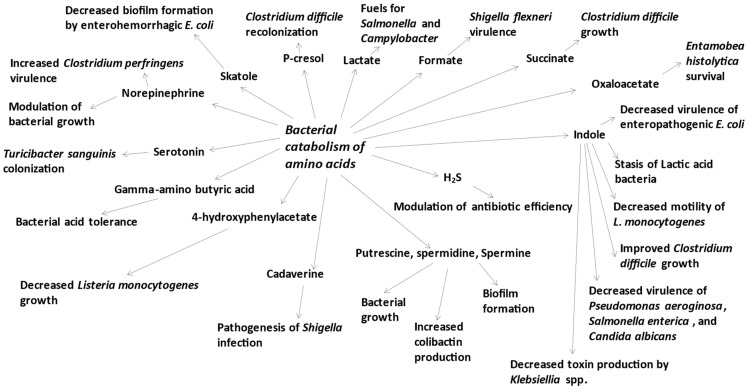
Schematic representation of the production of bacterial metabolites from amino acid precursors, and effects of these metabolites on microorganism biology.

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
