# Peer review of "Amino Acid-Derived Bacterial Metabolites in the Colorectal Luminal Fluid: Effects on Microbial Communication, Metabolism, Physiology, and Growth"

_microorganisms, 2023, doi:10.3390/microorganisms11051317_

Round 1

Reviewer 1 Report

Overall: a very nicely written and worthwhile review. Most main aspects of gut bacterial amino acid metabolism are covered in a comprehensive but, relatively parsimonious way (with a few caveats - see specific comments). I have a few specific requests related to referencing, which could be improved in some places, and the additional of minor amounts of important information. I especially enjoyed the integration and analysis of results, as well as research limitations, in the field to develop conclusions and generate some solid speculations - all too uncommon with reviews that tend towards a simple regurgitation information.

Specific Points

1. Referencing: while generally good (and better than most reviews I edit) there are still a number of reviews referenced where primary research papers should be. The following review references I noted should be replaced/added to with the primary research paper that first determined the information being provided and/or was foundational in establishing the scientific evidence: 14, 17, 26, 31, 39, 43, 49, 53, 54, 55 - there are others I may have missed, please ensure the use of reviews adheres to my following points. The author will know better than I what primary references to use but the general point is that reviews should be cited for information specifically only when the point being made is so general/broad it would require an unmanageable amount of references or if the information is peripheral to the main topic of the writing. The reviews the author cites can still be used - there is nothing with wrong with citing a good review that summarizes a finding well - but they should not exclusively be so.

2. A few further minor notes on referencing: reference 25 is a poor reference to cite on the general point of transmembrane proteins in bacteria as the source of amino acid acquisition or export. The review is now old and out-of-date (it was excellent no doubt at the time) and is not specifically devoted to discussing bacterial amino acid transporters. This is such a general topic would hesitate to recommend specific reviews, but the following can replace Saier's 2000 review as they are specific for bacterial amino acid transporters, and recent and are written by acknowledge leaders in this research field: https://doi.org/10.3389/fmolb.2021.699222, https://doi.org/10.1080/21505594.2016.1221025,  DOI:10.1007/7171_2006_069

while not all bacterial amino acid transporter families are covered in each of these reviews, they do overall cover the whole field and are far more comprehensive and up-to-date than the review provided. Please replace.

3. Referencing again: The introductory paragraph is excellent and insightful. One though I had when reading it: how well does this evidence correspond to the now classic evidence gathered from 1950-1980 on amino acid/peptide absorption in the gut (protein digestion). There are many references and a detailed description provided in the following work compiled in 1991 by Matthews: Protein absorption : development and present state of the subject | Semantic Scholar

I could not find a free copy online for you to access but as I read my copy, the evidence for the proportion of protein absorbed corresponding very well to the 90% figure you provide and the other evidence for the amount of nitrogenous material passing into the large intestine. Indeed, Matthews cites some the same papers you have here. I wonder if it would be worth consulting this text (the best I know of) and examining in detail if your figures are reliably reported in all the literature, perhaps even referencing some of the seminal paper (which you have missed) which provided the original evidence for the efficacy of protein absorption and nitrogen loads into the large intestine? One famous result report in detail in Matthews book but seemingly at odds with your introduction is that the majority of nitrogenous material measured in the large intestine is in the form of peptide or protein. Matthews details the discovery of the very famous result that peptides are in fact digested at a faster rate than single amino acids - so it would seem strange that, at least, a majority of nitrogenous products derived from alimentation would be peptides? Would the author like to comment on this? Or is there evidence that peptides present in the large intestine are endogenously produced and not primarily from alimentation?

4. One small section missing is a mention of the specifics of enzymatic digestion and absorption of peptides and amino acids in the host intestine (introduction). This fact is, of course, alluded to, but nowhere, that I can see, do you directly reference the specific mechanisms of protein digestion and absorption in the small intestine. This is of central importance to mention to your readers as the absorption of amino acids and peptide in the, mainly, small intestine but also large intestine (see below) plays a large role in both the availability of amino acids to gut bacteria and the potential absorption or otherwise of amino acid metabolites produced by gut bacteria. A small detailing of a few sentences is all that is required but important, along with appropriate references. The best general, current review references on amino acids digestions and absorption are: https://doi.org/10.1002/cphy.c170041, https://doi.org/10.1016/B978-0-12-809954-4.00047-5

please add a small overview and these references.

Another specific note on this topic that must also be include in brief is the subject of host amino acid absorption by the large intestine. For a long time disputed, the subject is given good treatment in these reviews: https://doi.org/10.3945/jn.117.248187, DOI: https://doi.org/10.1017/S0007114512002395, https://doi.org/10.3945/jn.117.248187, 

Specifically, the amino acid transporters ATB0+ (slc6a14 in Human genome annotation), PAT1 (slc36a1) and peptide transporter PEPT1 (slc16a1) have been shown to play a role in large intestinal amino acid/peptide absorption. This information should also be mentioned in a short additional section including these relevant references on large intestinal expression and their role and amino acid transport: https://doi.org/10.1152/ajpgi.00491.2012, https://doi.org/10.1113/jphysiol.2008.164228, https://doi.org/10.3389/fmolb.2021.646574, https://doi.org/10.1111/j.1476-5381.2012.02030.x

5. The short paragraph referenced by reference 17 (p. 4) seems a little undeveloped and out of place to me. Section 2 describes the uses of host-derived amino acids by the luminal bacteria, specifically for 'protein synthesis, energy production and other metabolic pathways ...' The short paragraph seems to be the sole allusion to the ' other metabolic pathways'. In particular, this short paragraph refers to detection of of AA-derived metabolites in fecal, urine, and colonic fluid. Where are the detail and primary references? Please provide. The other way i read this is as a short introductory/summary statement and that you (elsewhere in this section and the next) list the AA-derived metabolites referred to - if this is the case please make this clear.

6. Figure 1 is fine but please add the uptake AA pathways of the host lumen as described above i.e. ATB0+ (point 4 above). 

7. Figure 2: Is perhaps the only part of the review I found a little confusing (along with the text accompanying it, section 3). I this section and, hence, the fig. supposed to be comprehensive of all known functional effects of AA-derived metabolites by gut bacteria? If so, perhaps the Fig. 2 information is better presented as a table - complete with a column for references. Just a suggestion in this case.

8. No doubt section 3 metabolites are all able to be derived from AAs in the species from which the functional effect is reported (though TCA cycle anabolism and pathways specific to the particular bacterium). However, many of the metabolites you focus on can be derived in many cases from other host-derived alimentary sources e.g. carbohydrates and lipids. As one example, how is it known that the production of lactate (section 3.1) from the species listed is not, at least in significant part, simply a result of host alimentary hexoses via the bacterial pentose-phosphate, EMP, or other glycolytic pathways? Or as another example: my understanding is that formate is most likely produce in Shigella (and most gut anaerobes) from glucose via pyruvate oxidation to CO2 - is there any evidence for the production of formate and other metabolites you list (particularly those classically associate with anaerobic fermentation) from alimentary or endogenous amino acids? And if so to what extent compared with other sources? Perhaps a short paragraph somewhere in section 3 (the start) giving a general overview of this topic is in order; for no other reason than, like me, biochemists not specialized in this particular area will be confused by the very same points I am alluding to here? I have no doubt that in the many references you list this question (source of alimentary/endogenous precursors) is answered in the specific cases, but a general discussion, including a critical evaluation as you have conducted with other aspects of the field, may be in order. 

9. I find section 3.6 a little incomplete. It seems not all the evidence of polyamine function effects on large intestinal bacteria are detailed. I recently read the following review on this topic (https://doi.org/10.1093/bbb/zbac080) and find much information is missing. Naturally, the summary may be brief as it is a small part of this review, but I do ask that all main research findings are listed and summarized (and discussed if disputed or problematic).

10. I read several reviews late last year on a very similar topic and find that there are some alimentary AA-derived bacterial metabolites not discussed in section 3 e.g. kyneuric acid, tryptamin (see https://doi.org/10.1111/mmi.14905 & DOI: 10.2174/1389203721666200212095503 ).

In summary please, ensure that the review (particular section 3) is comprehensive in both information provided and primary papers referenced. It is very well written and would be a shame if it were not the comprehensive review that is suggested by the title. 

Reviewer 2 Report

Very complete review and appropriate to the theme of the special issue. A few small comments to improve it:

- Review the citations. They are not ordered and are difficult to follow.

- Figures are pointed out too often.

- In the introduction, at the end of the first paragraph, I do not understand why it is indicated. I wouldn't include it.
